# A tetravalent live attenuated dengue virus vaccine stimulates balanced immunity to multiple serotypes in humans

Usha K. Nivarthi [1,7], Jesica Swanstrom[2,7], Matthew J. Delacruz[1], Bhumi Patel[1], Anna P. Durbin[3], Steve S. Whitehead [4], Beth D. Kirkpatrick[5], Kristen K. Pierce[5], Sean A. Diehl [5], Leah Katzelnick [6], Ralph S. Baric [1,2✉] & Aravinda M. de Silva[1✉]

The four-dengue virus (DENV) serotypes infect several hundred million people annually. For the greatest safety and efficacy, tetravalent DENV vaccines are designed to stimulate balanced protective immunity to all four serotypes. However, this has been difficult to achieve. Clinical trials with a leading vaccine demonstrated that unbalanced replication and immunodominance of one vaccine component over others can lead to low efficacy and vaccine enhanced severe disease. The Laboratory of Infectious Diseases at the National Institutes of Health has developed a live attenuated tetravalent DENV vaccine (TV003), which is currently being tested in phase 3 clinical trials. Here we report, our study to determine if TV003 stimulate balanced and serotype-specific (TS) neutralizing antibody (nAb) responses to each serotype. Serum samples from twenty-one dengue-naive individuals participated under study protocol CIR287 (ClinicalTrials.gov NCT02021968) are analyzed 6 months after vaccination. Most subjects (76%) develop TS nAbs to 3 or 4 DENV serotypes, indicating immunity is induced by each vaccine component. Vaccine-induced TS nAbs map to epitopes known to be targets of nAbs in people infected with wild type DENVs. Following challenge with a partially attenuated strain of DENV2, all 21 subjects are protected from the efficacy endpoints. However, some vaccinated individuals develop post challenge nAb boost, while others mount post-challenge antibody responses that are consistent with sterilizing immunity. TV003 vaccine induced DENV2 TS nAbs are associated with sterilizing immunity. Our results indicate that nAbs to TS epitopes on each serotype may be a better correlate than total levels of nAbs currently used for guiding DENV vaccine development.

[1] Department of Microbiology and Immunology, University of North Carolina School of Medicine, Chapel Hill, NC, USA. [2] Department of Epidemiology, University of North Carolina Gillings School of Public Health, Chapel Hill, NC, USA. [3] Johns Hopkins Bloomberg School of Public Health, Department of International Health, Baltimore, MD, USA. [4] Laboratory of Infectious Diseases, NIAID, National Institutes of Health, Bethesda, MD, USA. [5] Vaccine Testing Center, Department of Microbiology and Molecular Genetics, University of Vermont Larner College of Medicine, Burlington, VT, USA. [6] Laboratory of infectious Diseases, NIAID, Bethesda, MD, USA. [7] These authors contributed equally: Usha K. Nivarthi, Jesica Swanstrom. ✉email: rbaric@email.unc.edu; aravinda_desilva@med.unc.edu

The four-dengue virus serotypes (DENV1-4) transmitted by mosquitos are estimated to infect several hundred million people each year living in tropical and sub-tropical regions around the world[1–3]. A primary infection with a DENV serotype results in long-term homotypic immunity (immunity against the serotype responsible for infection) and only transient cross protection against other serotypes[4,5]. Following primary infection, serotype-specific nAbs, which circulate for decades if not longer, are thought to mediate protection against the homologous serotype[6,7]. A person experiencing a second DENV infection with a different serotype faces a greater risk of developing severe dengue hemorrhagic fever and shock syndrome. The ability of some dengue-specific Abs to promote the entry of the virus into Fc receptor-bearing target cells is widely supported as the initiating event that culminate in severe disease[8,9]. Because of the possibility of a monovalent DENV vaccine increasing the risk of severe disease caused by heterologous DENV serotypes, all leading vaccines are based on tetravalent formulations designed to induce simultaneous and balanced protective immunity to all four serotypes. In this study, we use a human tetravalent DENV vaccination and DENV2 challenge model to characterize properties of Abs induced by a live-attenuated tetravalent DENV vaccine and to identify plausible correlates of protective immunity[10,11].

Several DENV vaccines are under development, including two live-attenuated tetravalent DENV vaccines currently in phase III clinical trials and one live-attenuated tetravalent vaccine, Dengvaxia, developed by Sanofi Pasture that has been licensed for use in children with pre-existing immunity to DENV[12–14]. In children with no immunity to DENVs at baseline, Dengvaxia was poorly efficacious and the vaccine increased the risk of severe dengue disease upon exposure to wild type DENV infections. Dengvaxia performed poorly in this population even though the vaccine-induced nAbs to the four serotypes. While nAbs have been considered to be a good correlate of protective immunity for flavivirus vaccine development, the Dengvaxia experience indicates otherwise for DENV vaccines. An unbalanced replication of vaccine components was observed in Dengvaxia. The DENV4 component was replication and immunodominant compared to the other three serotypes[15–20]. Moreover, unlike the DENV1, 2 and 3 components, the DENV4 component of Dengvaxia elicited serotype-specific nAbs in majority of dengue-naive individuals[21]. In Dengvaxia clinical trials, vaccine efficacy was highest against DENV4[14,22]. These observations indicate that independent replication of each vaccine component above a threshold is required for protection. In the current study, we analyze immune serum samples from subjects who received a live-attenuated tetravalent DENV vaccine to test if DENV serotype-specific Abs, which are linked to the balanced replication and immunogenicity of each vaccine component are elicited by the vaccine.

The Laboratory of Infectious diseases at the National Institutes of Health has attenuated DENVs by removal of 30 nucleotides from the 3′ untranslated region (UTR) and developed a live-attenuated tetravalent (rDENV1Δ30, rDENV2/4Δ30, rDENV3Δ30/31 and rDENV4Δ30) DENV vaccine designated TV003 for clinical testing in humans[23–28]. Here, we analyze the fine specificity of TV003-induced Abs to determine if each vaccine component is capable of independently stimulating immunity. In addition, the vaccinated subjects were also challenged with a partially attenuated strain of DENV2 to determine if the binding specificity of vaccine-induced nAbs is a better correlate of protection than the total level of nAbs currently used for vaccine evaluation.

## Results

### Level and serotype specificity of DENV nAbs induced by TV003.

Sera from 21 individuals were tested 6 months after receiving a single dose of the NIH live-attenuated tetravalent DENV vaccine (TV003) to measure the levels of total and serotype-specific nAbs induced by vaccination. The vaccine stimulated nAbs ($nAb_{50}$ titer >40) to all four serotypes in 62% (13/21) of subjects and to two or three serotypes in the remaining subjects (Supplementary Fig. 1 and Supplementary Table 1). The presence of nAbs alone to a particular serotype does not establish replication and immunogenicity of a vaccine component because Abs stimulated by one vaccine serotype can cross neutralize other serotypes. To determine the independent contribution of each vaccine component to immunity, we measured levels of nAbs to unique type-specific (TS) epitopes on each serotype. As described previously[21,29], we used beads coated with DENV serotypes to deplete different populations of Abs before performing DENV binding and neutralization assays. A TS response is the fraction of Abs against a particular serotype that remained after using heterologous DENV serotypes to remove all cross-reactive Abs. For example, subject 6 had Abs that bound to all 4 serotypes (Fig. 1a). When the sample was depleted using a mix of DENV1, 3 and 4 antigens, we observed complete loss of DENV1, 3 and 4, but not DENV2 binding and nAbs indicating that the subject had DENV2 TS Abs (Fig. 1a, b). When the same sample was depleted with DENV2 antigen, ELISA confirmed loss of DENV2 Abs, while retaining DENV1, 3, and 4 binding and nAbs (Fig. 1a, b). These results indicate that all 4-vaccine components in TV003 replicated in subject 6 and independently stimulated immunity to each DENV serotype.

Using the Ab depletion method, we estimated levels of TS Abs in all 21 subjects vaccinated with TV003 (Fig. 2 and Supplementary Table 2). The total number of individuals who developed TS nAbs to each serotype varied (13/21 for DENV1, 16/21 for DENV2, 18/21 for DENV3 and 21/21 for DENV4) (Fig. 2a). The absolute mean titer of TS nAbs to DENV1 was lower than responses to the other three serotypes (Fig. 2b). Overall, 16/21 (76%) subjects had TS nAbs to 3 or 4 different DENV serotypes (trivalent or tetravalent) (Fig. 2c). We conclude that in most volunteers, a single dose of TV003 resulted in the replication of 3 or 4 vaccine components and the stimulation of independent immunity to 3 or 4 DENV serotypes.

### Mapping the epitopes of TV003-induced DENV serotype-specific nAb responses.

Having established that TV003 stimulated TS nAbs to each DENV serotype, next we mapped epitopes on each DENV serotype recognized by these Abs. We have previously defined major epitopes on each DENV serotype recognized by TS nAbs that develop after WT DENV infections[30–33]. We have also developed epitope transplant recombinant DENVs for tracking serum-nAbs to TS epitopes on DENV1, 2, and 3[33–37]. Here we, first, describe a chimeric DENV recently created to map and track nAbs recognizing TS epitopes on DENV4[33]. Next, we use our panel of chimeric DENVs to determine if nAbs stimulated by TV003 track with known TS epitopes on each serotype.

### A recombinant DENV2/4 chimera for mapping DENV4-specific Ab responses.

DENV4-126 and DENV4-131 are DENV4 TS-neutralizing monoclonal Abs (mAbs) isolated from people exposed to DENV4 infections[33]. These Abs bind to quaternary structure epitopes on EDII and the hinge region between EDI and II[33]. To measure the levels of DENV4 TS Abs directed to this region, we constructed a recombinant chimeric DENV2 strain, where 26 amino acids in EDII and the EDI/II hinge region were changed to match residues on DENV4 (Fig. 3a, b). The rDENV2/4 virus, which replicated well in vero and C6/36 cells, was efficiently neutralized by DENV4-126 and DENV4-131, demonstrating structural and functional transplantation of epitopes recognized by

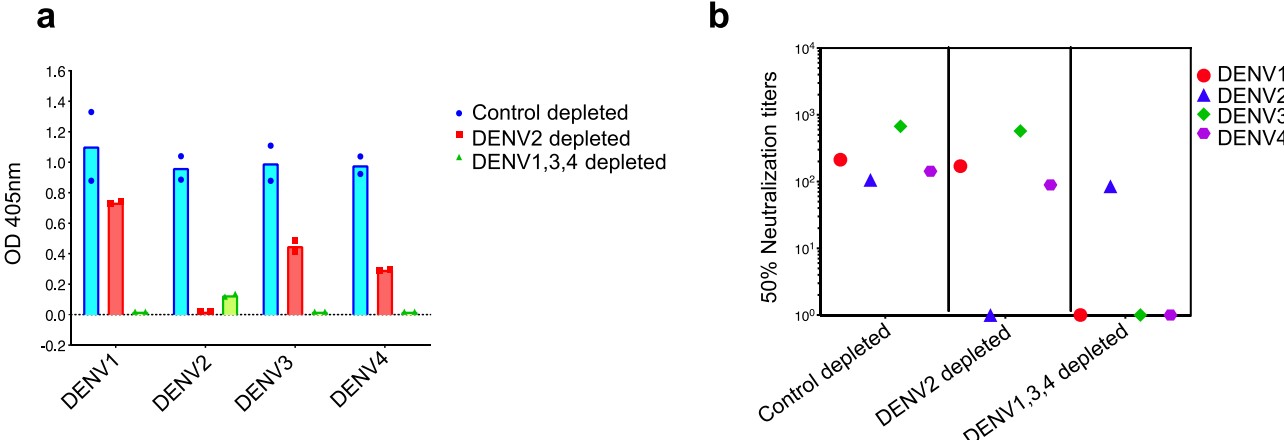

**Fig. 1 DENV antibody depletion assay to estimate proportions of serotype-specific and cross-reactive-neutralizing antibodies in a subject who received TV003.** Subject 6 was tested 6 months after vaccination to measure levels of TS binding and neutralizing antibodies to each DENV serotype. The sample was incubated with immobilized bovine serum albumin (control depleted), DENV2 antigen (DENV2 depleted) and a mix of DENV1, 3 and 4 antigens (DENV1, 3, 4 depleted) to remove specific populations of antibodies. The DENV2 depletion resulted in the removal of DENV cross-reactive and DENV2 TS antibodies, while retaining any DENV1, 3, or 4 TS antibodies in the sample. The DENV1, 3, 4 depletion resulted in the removal of DENV cross-reactive and DENV1, 3, and 4 TS antibodies, while retaining any DENV2 TS antibodies in the sample. The sample depleted of specific antibody populations was tested by **a** ELISA to detect any DENV1, 2, 3, or 4 type specific binding antibodies. Data was analyzed from subject 6 and represented as bars from a single independent experiment and expressed as mean ODs with corresponding data points shown. **b** Neutralization test to measure levels of TS-neutralizing antibodies to each DENV serotype. Data was analyzed from subject 6 and represented as dots from a single independent experiment and expressed as 50% neutralization titers.

these human mAbs. The DENV4 TS-neutralizing mAb 5H2[38], isolated from non-human primate infected with DENV4, which binds to an epitope outside the transplanted region, did not neutralize the chimeric virus. rDENV2/4 remained sensitive to neutralization by DENV serotype cross-reactive human mAbs EDE1 C8 and EDE2 B7[39] and DENV2 TS human mAbs 2D22[31], 3F9[35,36], and 4J23[40], which bind to regions that were not mutated in the chimera (Fig. 3c). Next, we used human immune-sera to assess if the DENV4-126 and DENV4-131 epitopes displayed on DENV2 were targets of serum nAbs that develop after primary DENV4 infections. Two of the three primary DENV4 sera had nAbs that tracked with rDENV2/4 (Fig. 3d). However, not all DENV4 nAbs in the two subjects tracked with the chimera indicating that other neutralizing epitopes are also targets of DENV4 TS Abs. The rDENV2/4 virus was efficiently neutralized by primary DENV2 immune sera demonstrating that major epitopes targeted by DENV2 TS Abs were preserved on the chimera (Fig. 3d). Our results demonstrate that the rDENV2/4 chimera displays DENV4-126 and 131 epitopes, while preserving the overall structural integrity of the DENV2 envelope outside the mutated region.

**Mapping TV003-induced DENV4 serotype-specific nAb response.** To map the DENV4 TS responses induced by TV003, we used the rDENV2/4 chimeric virus displaying the DENV4-126 and DENV4-131 epitopes on a DENV2 E protein backbone (Table S3 and Fig. 4a). As TV003 stimulates nAbs to multiple DENV serotypes, it was necessary to remove all Abs binding to the parental backbone virus (DENV2 in this case) before testing if the remaining DENV4 TS Abs tracked with the DENV4 epitope displayed on the chimera. We performed neutralization assays with DENV2, DENV4 and rDENV2/4 viruses and TV003 vaccine immune sera depleted of all DENV2-binding Abs. As expected, none of the subjects neutralized DENV2 after depletion of DENV2-binding Abs (Fig. 4b). Approximately half the sera (9/19) depleted of DENV2-binding Abs neutralized the rDENV2/4 chimera displaying DENV4-126 and 131 epitopes (Fig. 4b). As an alternate strategy for mapping DENV4 nAb responses, we performed neutralization assays with a rDENV4/3 virus displaying a

transplanted DENV3 TS nAb epitope (5J7) that spans the EDII and EDI/II hinge regions of E protein[37]. The 5J7 epitope transplant disrupts the epitopes recognized by human mAb DENV4-126 and 131 epitopes (Supplementary Fig. 2)[33]. Using 8 TV003 immune sera depleted of DENV3-binding Abs, we observed a significant drop in neutralization of the rDENV4/3 chimeric virus compared to WT DENV4 (Supplementary Fig. 2). Using both gain and loss of function assays, these results collectively, demonstrate that TS epitopes that encompass DENV4-126 and DENV4-131 epitopes are important but not exclusive targets of TV003 vaccine stimulated DENV4 nAbs.

**Mapping TV003-induced DENV1 serotype-specific nAb response.** To map the TV003-induced DENV1 nAb response, we used a chimeric rDENV2/1 virus displaying a major DENV1 TS epitope recognized by human mAb 1F4 (Fig. 5a and Table S3)[34] and immune sera from eight subjects who developed DENV1 TS nAbs after vaccination (Fig. 5). All eight subjects efficiently neutralized DENV1 after removal of all DENV2-binding Abs (DENV cross-reactive + DENV2 TS Abs), confirming that the vaccine stimulated DENV1 TS nAbs (Fig. 5b). As expected, after the DENV2 Ab depletion, none of the subjects neutralized DENV2. After removal of all DENV2-binding Abs, 6/8 subjects (75%) had Abs that neutralized the rDENV2/1 chimera demonstrating that TV003 stimulates DENV1 TS nAbs that bind to the region on EDI defined by mAb 1F4 (Fig. 5b).

**Mapping TV003-induced DENV3 serotype-specific nAb response.** To map the TV003-induced DENV3 Ab response, we used a chimeric rDENV4/3 virus displaying the DENV3 TS epitope recognized by human mAb 5J7 (Fig. 6a and Table S3)[34] and serum from eight subjects who had DENV3 TS nAbs after vaccination (Fig. 6). The immune sera were depleted of all DENV4-binding Abs before testing for Abs that neutralized the parental and rDENV4/3 chimeric virus (Fig. 6a). The sera depleted of all DENV4-binding Abs (DENV cross-reactive + DENV4 TS Abs) neutralized DENV3 but not DENV4 (Fig. 6b)

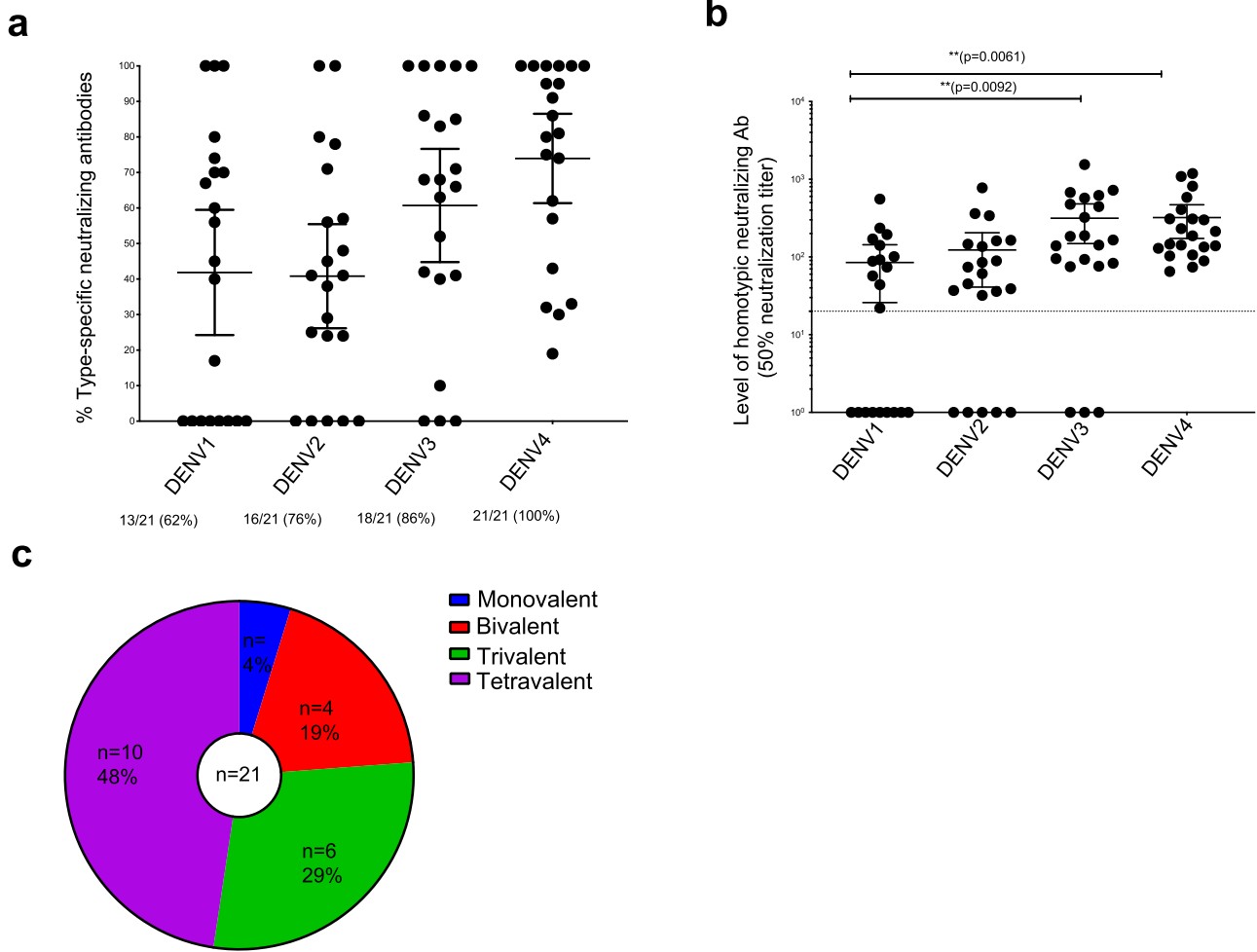

**Fig. 2 Proportion and level of homotypic-neutralizing antibodies to each serotype in DENV naive subjects who received TV003.** The antibody depletion assay was used to measure proportions of DENV TS (homotypic)-neutralizing antibodies induced by TV003 in ($n = 21$) DENV naive subjects. **a** The percentage of the total-neutralizing antibody response to each serotype contributed by the serotype-specific antibodies for each subject. **b** The absolute level of homotypic-neutralizing antibody to each serotype for each subject. **c** Number and % of subjects that developed an absolute TS nab titer >20 is depicted. Data were analyzed from a total number of 21 subjects indicated as dots from a single independent experiment and expressed as mean titers ± 95% confidence intervals (**a**, **b**). Nonparametric one-way Anova with Friedman Dunn's multiple comparisons test was used for determining the statistical differences between the level of TS nAbs across serotypes (**b**). Each p-value was adjusted to account for multiple comparisons. (****$p < 0.0001$; ***$p < 0.001$; **$p < 0.01$; *$p < 0.05$). The serotypes with significantly high levels of TS nabs compared to other serotypes is shown (**b**) (DENV3: **$p = 0.0092$; CI:148.4–481.5; DENV4: **$p = 0.0061$; CI: 172.9–469.8). The dotted line indicates the cut off for the absolute type specific titer.

confirming the presence of vaccine stimulated DENV3 TS nAbs. However, these nAbs did not efficiently track with DENV3 5J7 epitope displayed on the chimeric virus because only 2/8 samples (25%) showed any neutralization of the chimeric virus (Fig. 6b). We conclude that the 5J7 epitope is not a major target of TV003-induced DENV3 TS nAbs.

**Mapping TV003-induced DENV2 serotype-specific nAb response.** To map the TV003-induced DENV2 Ab response, we used a chimeric rDENV4/2 virus displaying the DENV2 TS epitopes centered on EDIII recognized by human mAb 2D22 and 1L12[31,35,36] (Table S3) (Fig. 7) and serum samples from 16 subjects, who received the vaccine and developed TS responses to DENV2. The samples were depleted with DENV4 antigen to remove all Abs binding to DENV4 E backbone of the rDENV4/2 chimera. Depleted sera from all but one subject (15/16) neutralized DENV2 (>90%) confirming the presence of DENV2 TS nAbs (Fig. 7b). In fourteen subjects, the TS nAbs tracked with EDIII of DENV2 displayed on the rDENV4/2 chimera (Fig. 7b).

We conclude that most subjects vaccinated with TV003 develop DENV2 TS nAbs that map to epitopes centered on EDIII, which is a major target of nAbs that develop after WT DENV2 infections.

**Properties of TV003-induced DENV2 nAbs correlated with sterilizing immunity.** As reported previously, when the 21 subjects who received TV003 were challenged 6 months later with DENV2, none of the subjects had detectable virus in the serum or developed the rash often observed in people infected with this strain of DENV2[11]. Unlike the previously reported method of calculating the boost as a fourfold rise in nAb titer against DENV2 by study day 270[11], we independently calculated boost and to be more stringent, we estimated boost as twofold rise in nAb titer against DENV2 by day 208. In five individuals, we observed that the DENV2 challenge stimulated a boost (≥2-fold) in DENV2 nAbs at day 28 post-infection, while no boost was observed in the other 16 subjects. Four of the 5 subjects with an Ab boost also developed dengue-specific IgM Abs following

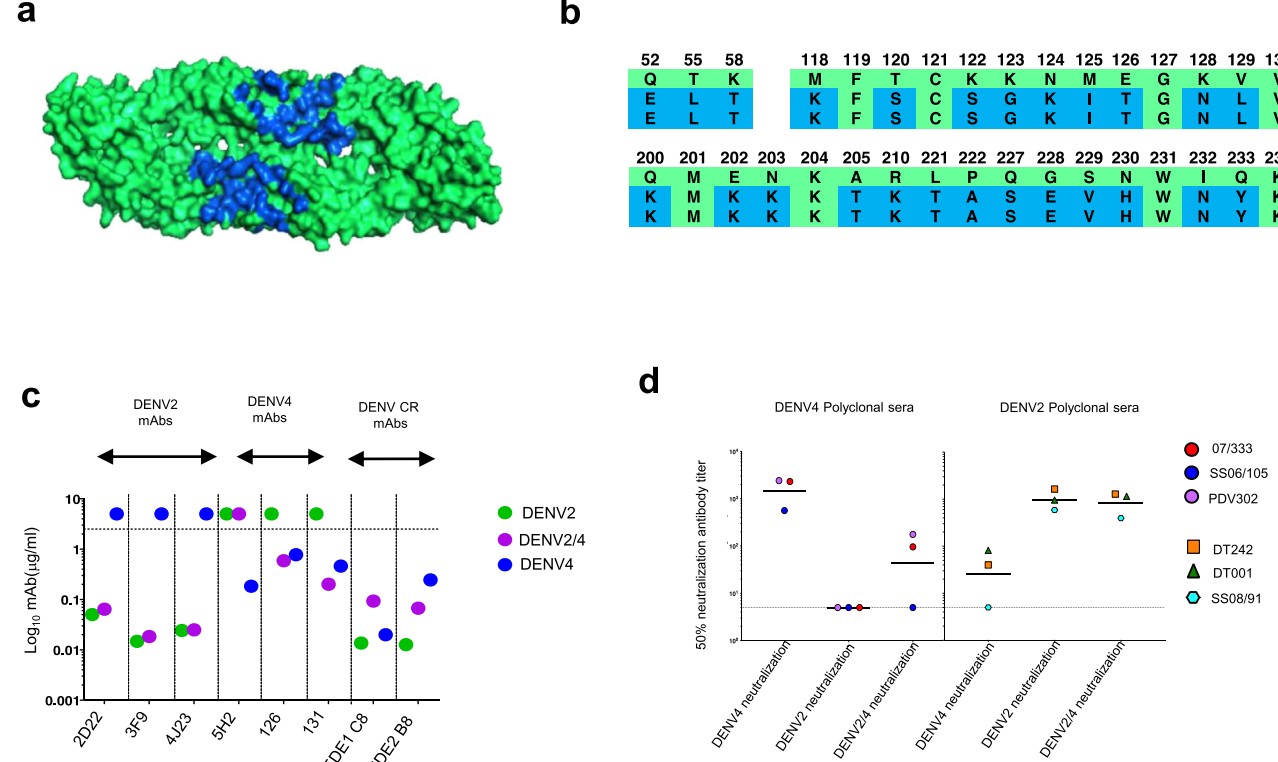

**Fig. 3 Characterization of DENV4 126 and 131 epitope transplant rDENV (rDENV2/4) used for mapping the DENV4 responses. a** Recombinant DENV2 containing EDI/II hinge, EDII residues from DENV4 that includes epitopes of the DENV4-126 and DENV4-131 mAbs. The figure depicts a model of DENV2 E protein dimer, with the mutated residues colored in blue. **b** Amino acid alignment of DENV2, rDENV2/4, DENV4, protein sequences in the EDI/II hinge; EDII regions with mutated residues in rDENV2/4 colored in blue. **c** Monoclonal antibodies (2D22, 3F9, 4J23: DENV2 TS mAbs, 5H2, 126,131: DENV4 TS mAbs, EDE1 C8, EDE2 B7 pan DENV specific mAbs) neutralization of rDENV2/4 compared to each parental strain's is shown. Data shown here was from a single independent experiment and represented as mean neutralization titers. **d** Characterizing the neutralizing properties of DENV2 ($n = 3$) and DENV4 ($n = 3$) polyclonal natural infection sera against rDENV2/4 and the parental viruses. Data were obtained from a single independent experiment and expressed as mean titers.

challenge, whereas as an IgM response was not observed in people with no Ab boost after challenge (Table S4). These results demonstrate likely sterilizing DENV2 immunity induced by TV003 in most subjects (no boost) and low level, replication of challenge virus, leading to a DENV2 Ab boost and DENV-specific IgM in other subjects (Table S4). We performed an immune correlates analysis to identify properties of Abs induced by TV003 associated with the transient post-challenge ≥2-fold rise in DENV2-specific Ab titers (Table 1 and Supplementary Fig. 3). We considered as predictors the pre-challenge geometric mean titer (GMT) of nAbs to the 4 DENV serotypes, titer of total DENV2 nAbs, titer of TS DENV2 nAbs, and the percentage of DENV2 TS nAbs. We evaluated whether the pre-challenge titer correlated with fold-change in Ab boost (Pearson's product-moment correlation), as well as whether some threshold of pre-challenge titer was associated with reduced odds of a boost in DENV2 nAb titer of ≥2-fold (expressed as an odds ratio, presented with 95% confidence intervals). We tested a range of threshold values corresponding to deciles of each predictor variable and evaluated the significance of the association with ≥2-fold nAb boost. For vaccine-induced GMT of nAbs to the 4 serotypes or total levels of DENV2 nAbs, we did not observe a correlation or any threshold value that resulted in a significant association with nAb boost after challenge (Table 1 and Supplementary Fig. 3). In contrast, detectable levels of TS DENV2 nAbs, measured both as DENV2 TS nAbs and percentage of DENV2 TS nAbs, were associated with significant reduction in the odds of boost after challenge. Further, percentage of DENV2 TS nAbs was

significantly correlated with the magnitude of the post-challenge boost (Table 1 and Supplementary Fig. 3). These results suggest that DENV TS nAb is a better predictor of independent immunity stimulated by each vaccine component and serotype-specific vaccine efficacy than total level of nAbs, widely used as a correlate for dengue vaccine development.

## Discussion

Three live-attenuated tetravalent DENV vaccines are at different stages of advanced clinical development[11,14,41–43]. Dengue vaccine developers have relied on nAb as a correlate of protection because nAbs, even at low levels, were correlated with the efficacy of other flavivirus vaccines like yellow fever vaccine (YFV), Japanese encephalitis vaccine (JEV), tick-borne encephalitis (TBEV). Recent results from a dengue vaccine clinical trial challenge this assumption because many children experienced breakthrough infections even though the vaccine stimulated nAbs[14]. Population level, pooled analysis of data from dengue seronegative and seropositive children who received the vaccine demonstrate that children with high titers of nAbs were more likely to be protected than children with low titers[44]. However, when the data are stratified to analyze vaccine responses in children who were dengue seronegative (naive) at baseline, total nAbs to each serotype are poorly correlated with vaccine efficacy[14]. The goal of the current study was to perform a detailed analysis of the properties of nAbs stimulated by a tetravalent live-attenuated DENV vaccine in seronegative individuals to identify

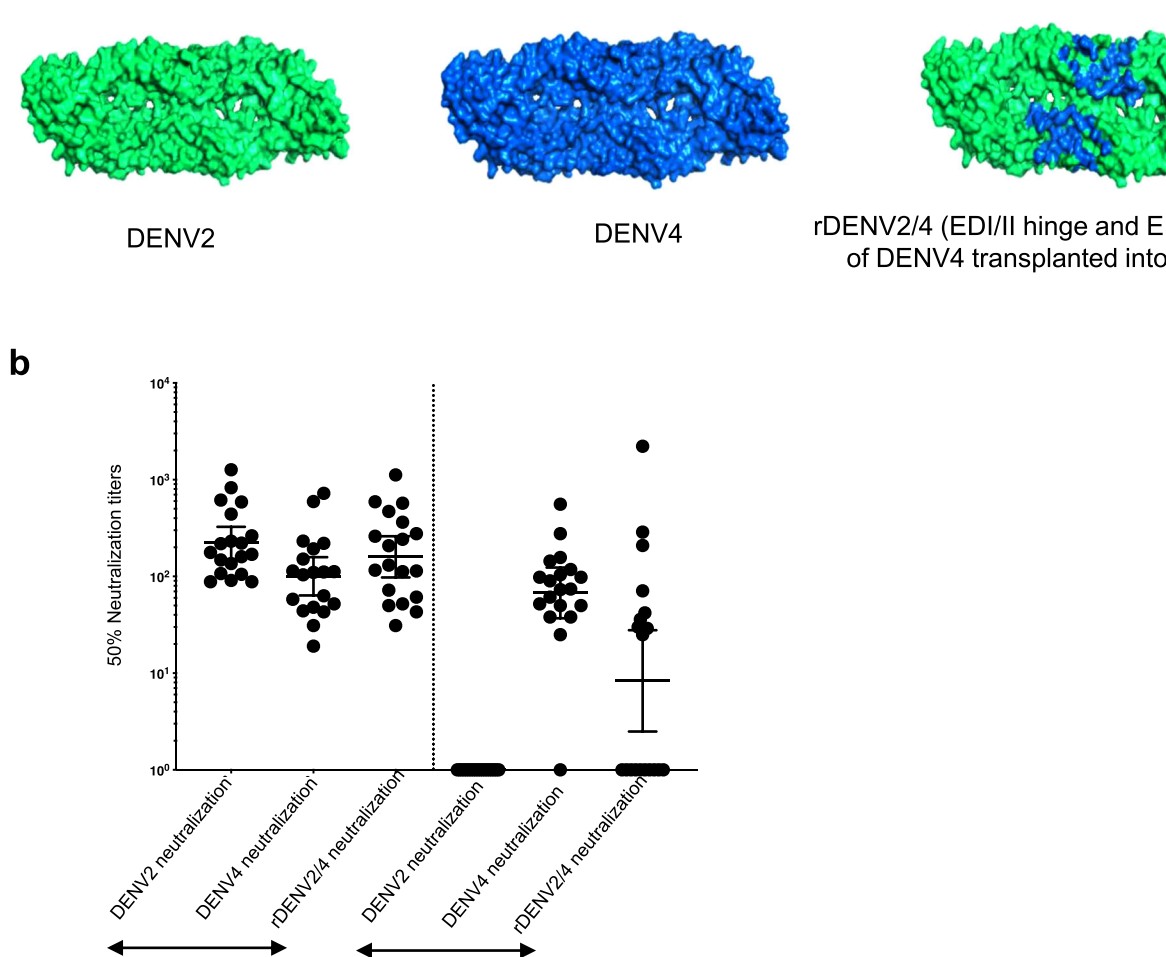

**Fig. 4 TV003 induce varying levels of DENV4 serotype-specific responses against known DENV4-neutralizing epitopes. a** DENV2 wt (green), DENV4 wt (blue), rDENV2/4 with DENV4-specific 126, 131 mAb epitopes transplanted into DENV2 (shown in blue) are depicted. **b** Panel of 19 sera (*n* = 19) with evidence of DENV4 TS responses were subjected to depletions against DENV2 to map the DENV4 TS nAb epitopes using the rDENV2/4 virus. This is a gain of function virus. Figure shows the 50% neutralization antibody titers for the BSA depleted and the DENV2 depleted serum samples against DENV2 wt, DENV4 wt and the rDENV2/4 viruses. Data were analyzed from a total number of 19 subjects indicated as dots from a single independent experiment and expressed as geometric mean titers ± 95% confidence intervals of the grouped samples (**b**).

Ab profiles that are more predictive of outcome than the current standard.

While Dengvaxia was developed to be a vaccine with four live chimeric viruses independently replicating and stimulating immunity to each DENV serotype, the DENV4 component of the vaccine replicates better[16] and stimulates higher levels of nAbs compared to the other three components. In fact, in many DENV seronegative individuals, the vaccine acts like a monovalent DENV4 vaccine[21,29]. Similarly, another tetravalent live-attenuated DENV vaccine developed by Takeda Vaccines, which is currently being evaluated in clinical trials, induces higher levels of nAbs to DENV2 compared to the other three serotypes[45]. In the current study, we used a human vaccination and challenge model to dissect the properties and performance of the NIH TV003 tetravalent live-attenuated DENV vaccine. Our initial goal was to measure levels of vaccine-induced nAbs directed to TS (unique) epitopes on each DENV serotype. Even if TS Abs are not directly responsible for protective immunity, we reasoned that Abs directed to unique epitopes on each serotype are a better indicator of the replication and immunogenicity of each vaccine component compared to total Abs directed to epitopes that are unique and conserved across serotypes. In previous study, we observed each monovalent component of the NIH vaccine administered to people induced TS nAbs to epitopes known to be targeted by nAbs in people infected with wild type DENVs[46]. In the current study, we observe a similar immunogenicity profile when all four components are delivered together as a single-dose tetravalent vaccine indicating that interference and immunodominance are not major problems for TV003. A similar analysis we recently completed for Dengvaxia demonstrated that most people develop high levels of TS nAbs DENV4 only, which is consistent with the replication and immunodominance of DENV4 in this vaccine[21].

The 21 subjects immunized with TV003 were challenged 6 months later with a partially attenuated strain of DENV2. As reported previously, all the subjects were protected from DENV2 viremia, rash and neutropenia[11]. Therefore, these primary virologic and clinical outcomes could be used to identify Ab responses correlated with efficacy. We observed no boosting of DENV2 nAbs in most vaccinated individuals after DENV2 challenge, which is consistent with TV003 inducing sterilizing immunity against DENV2. In a few individuals, we observed boosting of DENV2 nAbs and an increase in IgM indicating some

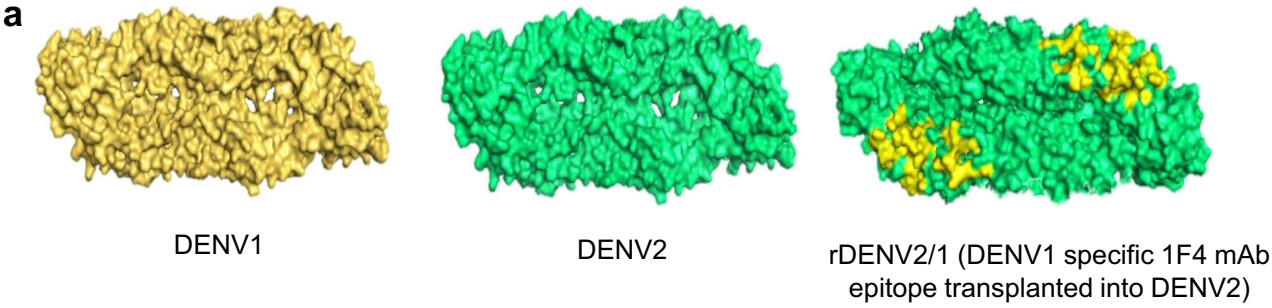

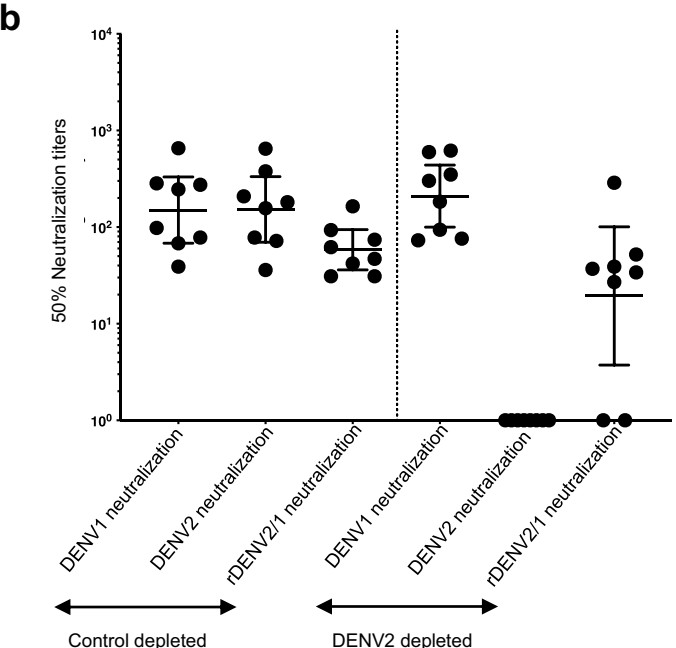

**Fig. 5 TV003 induce varying levels of DENV1 serotype-specific responses against a known DENV1-neutralizing epitope. a** DENV1wt (yellow), DENV2wt (green), rDENV2/1 with DENV1-specific 1F4 mAb epitope transplanted into DENV2 (shown in yellow) are depicted. **b** Panel of 8 TV003 sera (*n* = 8) with evidence of DENV1 TS responses were subjected to depletions against DENV2 to map the DENV1 TS nAb epitopes using the rDENV2/1 virus. Figure shows the 50% nAb titers for the BSA (control) depleted and the DENV2 depleted serum samples against DENV1 wt, DENV2 wt and the rDENV2/1 viruses. Data were analyzed from a total number of eight subjects indicated as dots from a single independent experiment and expressed as geometric mean titers ± 95% confidence intervals of the grouped samples (**b**).

replication of the challenge virus in these individuals. In an immune correlate's analysis using Ab boosting as the outcome, we observed a correlation between the presence of DENV2 TS nAbs and the post-challenge Ab boosting. We did not observe any correlations between total levels of DENV2 nAbs and post challenge Ab boosting. While these observations must be interpreted with caution, due to the small sample size and overall protective efficacy of TV003 against challenge, they support the hypothesis that in seronegative individuals who are vaccinated, TS nAbs are a better correlate of dengue vaccine efficacy than total levels of nAbs that can be derived from the replication of homologous and heterologous vaccine components.

## Methods

### Study design, subjects, and sera

*Study design of the clinical trial.* All sera tested in these experiments came from healthy volunteers enrolled in a trial designed by the NIH to assess the protective efficacy of TV003 against DENV2. This is a randomized, double-blind, placebo-controlled trial[11]. The studies were performed under an investigational new drug application reviewed by the U.S. Food and Drug Administration and approved by the Institutional review Boards at the University of Vermont (UVM) and Johns Hopkins University (JHU)[11]. Informed consent was obtained in accordance with federal and international regulations (21CFR50 and ICHE6). Independent monitoring was performed externally and the National Institute of Allergy and Infectious Diseases safety monitoring board reviewed all safety data every 6 months. A total of 48 subjects were enrolled under study protocol CIR287 (ClinicalTrials.gov NCT02021968) between 11 November 2013 and 25 February 2014. The entire detailed study or the clinical protocol is published previously[11]. Two groups of 24 subjects (12 TV003 recipients and 12 placebo recipients) were tested at sites in Baltimore, MD and Burlington, VT. The primary efficacy endpoint of the study was the protection afforded by the vaccine against viremia induced by the challenge virus rDENV2Δ30. The study was powered to detect a protective efficacy against viremia of 60% at $\alpha = 0.05$ with 80% power[11]. Secondary endpoints included protection against rash and neutropenia. At day 0, 24 randomized subjects received a single subcutaneous dose of TV003 ($10^3$ PFU of TV003), and 24 randomized subjects received a placebo control (L-15 vaccine diluent). Six months later, subjects were challenged with $10^3$ PFU of rDENV2Δ30 by subcutaneous injection. Forty-one subjects returned for challenge (21 TV003 recipients and 20 placebo recipients). The most common adverse event (AE) after TV003 administration was a mild, asymptomatic rash that correlated with seroconversion to all four DENV serotypes. Oral temperatures were recorded three times daily for 16 days. Clinical assessments and physical examinations were performed every other day through day 16 and on days 21, 28, 56, 150, and 180[23]. The schedule for visits after challenge was same except study day 150 was not performed. AEs and clinical results were recorded and graded for intensity. Infection with TV003 or rDENV2Δ30 was defined as recovery of virus from the blood and /or seroconversion to DENV, as analyzed by plaque reduction neutralization assay (PRNT).

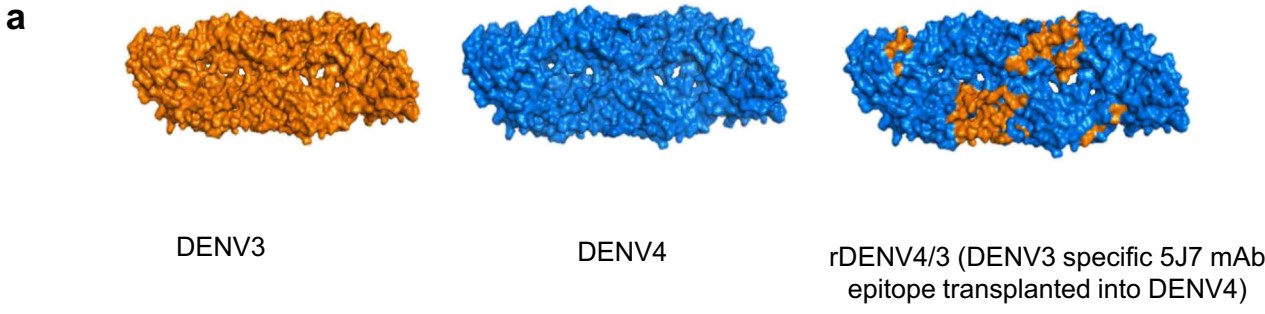

DENV3          DENV4          rDENV4/3 (DENV3 specific 5J7 mAb
                              epitope transplanted into DENV4)

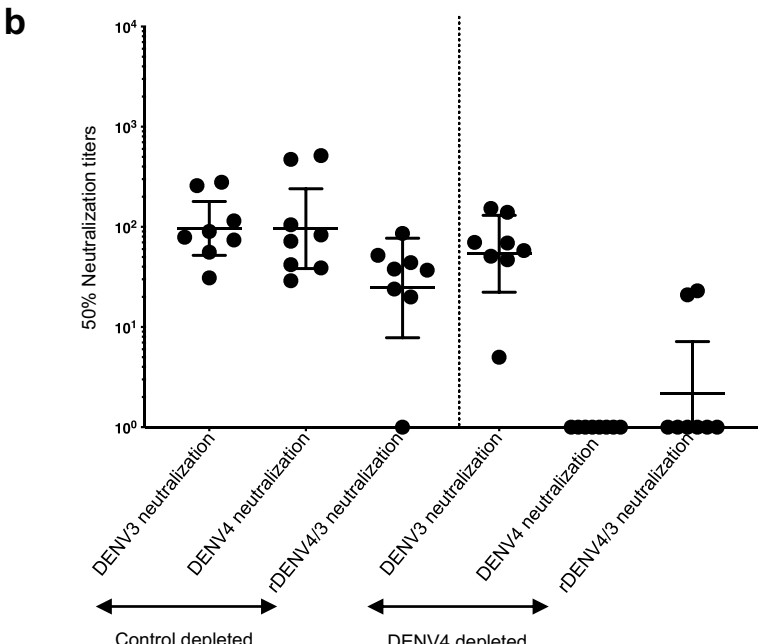

**Fig. 6 TV003 induce varying levels of DENV3 serotype-specific responses that do not track with known DENV3-neutralizing epitope. a** DENV3 wt (orange), DENV4 wt (blue), rDENV4/3 with DENV3-specific 5J7 mAb epitope transplanted into DENV4 (shown in orange) are depicted. **b** Panel of 8 TV003 sera ($n = 8$) with evidence of DENV3 TS responses were subjected to depletions against DENV4 to map the DENV3 TS nAb epitopes using the rDENV4/3 virus. Figure shows the Neut$_{50}$ titers for the BSA depleted and the DENV4 depleted serum samples against DENV3 wt, DENV4 wt and the rDENV4/3 viruses. Data were analyzed from a total number of eight subjects indicated as dots from a single independent experiment and expressed as geometric mean titers ± 95% confidence intervals of the grouped samples (**b**).

**Pre-specified outcomes reported from the clinical trial listed as primary and secondary objectives**. Protection of TV003 against viremia induced by rDENV2Δ30 (primary objective # 1 from the original protocol): the incidence, magnitude and duration of challenge virus viremia at study day 180 in subjects who received TV003 was compared to those who received placebo on study Day 0. All subjects vaccinated were protected from viremia induced by rDENV2Δ30[11].

Protective efficacy against rash and neutropenia (primary objective #2 from the original protocol): The proportion of subjects who received TV003 and developed rash/neutropenia following challenge were compared to the proportion of subjects who received placebo and developed rash or neutropenia following challenge. All subjects vaccinated were protected against rash and neutropenia[11].

Evaluate if inoculation with rDEN2Δ30 at 6 months will boost neutralizing antibody titers to DENV-1, DENV-2, DENV-3, and DENV-4 in those subjects who received TV003 on study Day 0 (Secondary objective #7 from the original protocol): Boost was defined as (≥4-fold rise in antibody titer) after challenge as measured by study day 270 compared with study day 180[11].

Sterilizing immunity induced by vaccination was defined as the prevention of rDENV2Δ30 infection as shown by a lack of viremia, rash, neutropenia, or antibody boost.

**Sera used in the current study**. Serum samples that were collected 6 months after vaccination with TV003 (21 subjects), or placebo (4 subjects) but before challenge, were analyzed in the current study in neutralization and depletion studies. The antibody depletion studies were performed to assess if TS nAbs were stimulated by each vaccine component and the epitope mapping studies were performed to assess

if the vaccine-induced TS nAbs target epitopes of previously mapped nAbs in natural infections. Finally, the correlation between the pre-challenge TS DENV2 nAbs and the fold rise in boost was assessed. We defined boost as ≥2-fold rise in serum neutralizing antibody titer by study day 208 compared to study day 180.

**Vaccine (TV003) strains**. TV003 is an admixture composed of three DENVs attenuated by deletion(s) in the 3′ untranslated region (3′ UTR): rDENV1Δ30, rDENV3Δ30/31, and rDENV4Δ30, and a fourth component that is a chimeric virus with the prM and E proteins of DENV2 NGC (New Guinea C strain) exchanged for DENV4 in the rDEN4Δ30 genome (rDENV2/4Δ30). L-15 medium (Cambrex Bioscience) was used to dilute the vaccine viruses immediately prior to vaccination. Vaccine virus titers were determined using a standard plaque assay[47].

**Viruses, cells**. DENV1 (American genotype; strain West Pac74), DENV2 (Asian genotype; strain S-16803), DENV3 (Asian genotype; strain CH-53489), and DENV4 (American genotype; strain TVP-376) (provided by Robert Putnak, Walter Reed Army Institute of Research, Silver Spring, MD) were used for both binding enzyme-linked immunosorbent assays (ELISAs) and neutralization assays unless otherwise noted. Purified antigens of the same aforementioned viruses were used in depletion studies. All viruses used in the neutralization assays were grown in C6/36 *Aedes albopictus* mosquito cells (American type Culture Collection; CRL-1660) and maintained in minimal essential medium (MEM; Gibco) at 32 °C. Vero-81 mammalian cells were used to generate purified antigens of the aforementioned DENV1-4 serotypes. Vero cells (American Type Culture Collection; CCL-81) were

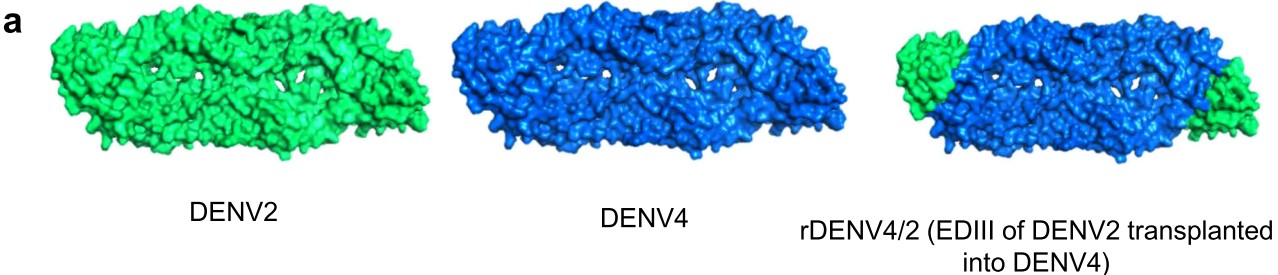

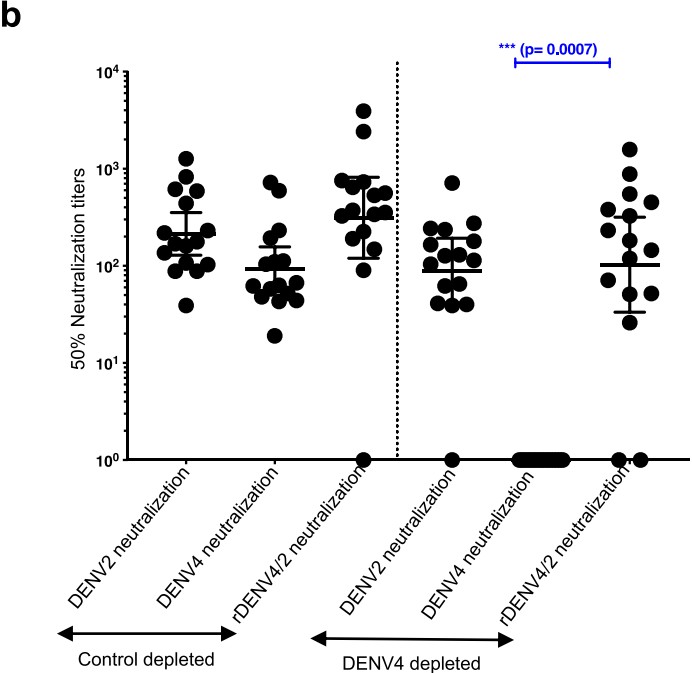

**Fig. 7 TV003 induce varying levels of DENV2 serotype-specific responses against known DENV2-neutralizing epitope domain. a** DENV2 wt (green), DENV4 wt (blue) rDENV4/2 with the entire DENV2 EDIII domain transplanted into DENV4 (shown in green) are depicted. **b** Panel of 16 TV003 sera ($n = 16$) with evidence of DENV2 TS responses were subjected to depletions against DENV4 to map the DENV2 TS nAb epitopes using the rDENV4/2 virus. Figure shows the Neut$_{50}$ titers for the BSA depleted and the DENV4 depleted serum samples against DENV2 wt, DENV4 wt and the rDENV4/2 viruses. Data were analyzed from a total number of 16 subjects indicated as dots from a single independent experiment and expressed as geometric mean titers ± 95% confidence intervals of the grouped samples. Nonparametric one-way Anova with Friedman Dunn's multiple comparisons test (**b**) was used for the epitope mapping data to determine the loss or gain in neutralization to the recombinant chimeric viruses compared to the wild type parental backbone viruses. Each *p*-value was adjusted to account for multiple comparisons. (****$p < 0.0001$; ***$p < 0.001$; **$p < 0.01$; *$p < 0.05$). The significant gain in neutralization of DENV2 TS nAbs against rDENV4/2 is indicated (***$p = 0.0007$; CI: 95.0–535.7).

**Table 1 Immune correlates analysis of TV003-induced antibodies and sterilizing immunity upon DENV2 challenge.**

| Predictor | Odds of experiencing a ≥2-fold boost in DENV2 nAb titer after challenge given predictor nAb levels above indicated threshold | | | | Correlation of pre-challenge nAb titer with boost in DENV2 nAb titer after challenge (Pearson's product-moment correlation) | |
|---|---|---|---|---|---|---|
| | Threshold titer of nAb | Odds % ratio | 95% CI | *p*-value | Correlation | *p*-value |
| Geometric mean of nAbs to all four DENV serotypes | 138 | 0.4 | 0.05–3.12 | 0.382 | 0.15 | 0.515 |
| Total titer DENV2 nAb | 74 | 0.35 | 0.04–3.08 | 0.341 | −0.23 | 0.313 |
| Type-specific titer DENV2 nAb | 20 | **0.1** | **0.01–0.97*** | **0.047** | −0.39 | 0.079 |
| % of DENV2 Type-specific nAb | 0 | **0.1** | **0.01–0.97*** | **0.047** | **−0.44** | **0.046*** |

All 21 vaccinated subjects ($n = 21$) were challenged with a partially attenuated strain of DENV2 6 months after vaccination. Sterilizing immunity is defined as <2-fold increase (compared to pre-challenge levels) in DENV2 nAb at day 28 after challenge.
Asterisk indicates significant correlations are in bold.
% The odds of experiencing a ≥2-fold boost in DENV2 nAb titer post-challenge was estimated by predictor nAb level, thresholded by decile. The threshold for the decile with the most significant relationship with fold-boost is presented. The threshold corresponding to the 20th percentile had the lowest *p*-value for the predictors: Total DENV2 nAb, TS DENV2 nAb, and % of TS DENV2 nAb. The threshold corresponding to the 40th percentile was most significant for the predictor Geometric mean of nAb titers to all four DENV serotypes. Predictors, geometric mean of nAb titers to all four DENV serotypes and total DENV2 nAb were not associated with a significant protective effect for any threshold value tested (20th, 30th, 40th, 50th, 60th, 70th, 80th percentiles).
Data were analyzed by an exploratory statistical analysis to evaluate the relationship between pre-challenge nAb titer and boost in DENV2 nAb titer post-challenge. The correlation between predictor and outcome variables were evaluated by using pearson's product-moment correlation. The odds ratio analysis of ≥2-fold boost in DENV2 nAb titer for those with predictor nAb levels above vs. at or below a specified threshold were estimated us the mosaic package in R.

maintained in Dulbecco's modified Eagle's medium-F12 (DMEM-F12) at 37 °C. All growth and maintenance media used were supplemented with 5% fetal bovine serum (FBS), 100 U/mL penicillin, 100 mg/mL streptomycin, 0.1 mM non-essential amino acids (Gibco), and 2 mM glutamine. All cells were incubated in the presence of 5% CO₂. The 5% FBS was reduced to 2% to make infection medium for each cell line. Virus produced in the vero-81 cell culture media was purified by density gradient and ultracentrifugation[48]. NGC virus was used to perform the neutralization assay with 6-month post vaccination serum samples and 28 post challenge serum samples for nAb boost determination for the data presented in supplementary Table 4.

**Whole virus depletion of DENV-specific Abs from tetravalent vaccine immune sera.** Depletions for the serum samples were performed to undesrstand which sub-populations of nAbs mainly contribute to neutralization against each serotype[49]. Purified DENV was absorbed onto 4.5-μm-diameter Polybead polystyrene microspheres (Polysciences, Inc., Cat. # 17135-5) at a bead (microliters) to ligand (micrograms) ratio of 5:2. Polystyrene beads were washed three times with 0.1 M borate buffer (pH 8.5) and incubated with the relevant purified DENV overnight at room temperature (RT). Control beads were incubated overnight with an equivalent amount of bovine serum albumin (BSA). The control and virus-adsorbed beads were blocked with BSA (10 mg/mL)–borate buffer for 30 min at RT three times and washed four times with phosphate-buffered saline (PBS). DENV-specific Abs were depleted from human tetravalent vaccine sera by incubating the virus-adsorbed beads with human sera diluted 1:10 in 1X PBS for 1 h at 37 °C with end-over-end mixing. Samples were subjected to at least three sequential rounds of depletions. Successful removal of the respective Abs was confirmed by ELISA. The ability of the depleted samples to neutralize viruses of all of the four serotypes was tested with a focus-forming assay in Vero-81 cells.

For the sera obtained at UVM, we used a more improved depletion assay that we are currently using as a standard way of performing depletions. This improved version yielded similar results as the previously used method of depletion with polystyrene beads. Briefly, Dynabeads™ M-280 tosylactivated were covalently bound to anti-DENV E mAb 1M7 (100 ug) overnight at 37 °C. Bead:mAb (5 mg:100 ug) complex was blocked with 1% BSA in PBS at 37 °C, and then washed with 0.1 M 2-(N-morpholino) ethane sulfonic acid (MES) buffer. Beads were incubated with BSA (control), purified antigens for 1 h at 37 °C, and then washed three times with PBS. Bead:mAb:DENV (5 mg:100 ug:100 ug) complex was fixed with 2% paraformaldehyde in PBS for 20 min, and then washed four times with PBS. DENV-specific antibodies were depleted from sera by incubating beads with sera diluted 1:10 in PBS for 1 h at 37 °C with end-over-end mixing for at least three sequential rounds of depletions. Removal of DENV antibodies was confirmed by DENV ELISA.

For all the sera, BSA coated beads were used for the control depleted condition to estimate the baseline neutralization titers before depletion. For estimating the DENV2 TS Abs, a condition in which beads adsorbed/conjugated with a mixture of DV1, 3, 4 was used. To estimate the TS Abs against DENV1, DENV3, DENV4, a condition in which beads adsorbed/conjugated with DENV2 was used. A protocol using a different depletion condition (e.g., DENV1 vs. DENV2,3,4 mix) to estimate DENV2, DENV3, DENV4 TS Abs and DENV1 TS Abs, respectively, would be expected to yield similar results as shown for two subjects in Supplementary Table 5.

**Enzyme-linked immunosorbent assays (ELISA).** Prior to depletion, titration ELISAs were done on the samples, where the ELISA plate was coated with purified DENV antigen to be used when depleting the sera. The initial ELISA helps quantify the amount of DENV antigen needed to fully deplete the sera of Abs against the respective serotype(s). After depletion, an ELISA was done to confirm that the sera did not contain Abs that could bind to the serotype used in depletion. Purified antigen was plated at 100 ng/well in 96-well ELISA plates overnight at 4 °C. Plates were blocked with 3% (vol/vol) normal goat serum (Gibco, Thermo Fisher, USA)-Tris buffered saline (TBS) −0.05% (vol/vol) Tween 20 (blocking buffer). Sera were diluted and added to the antigen coated ELISA plates. Alkaline phosphatase-conjugated secondary Abs (anti-human IgG-AP; Sigma) were used at a dilution of 1:2500 to detect binding of sera with p-nitrophenyl phosphate substrate, and reaction color changes were quantified by spectrophotometry at 405 nm.

**IgM MAC ELISA.** Day 180 post vaccination serum from each of the 21 subjects and the longitudinal sample, 28 days post challenge was analyzed in an IgM MAC ELISA. Anti-IgM was diluted 1:50 in 0.1 M Carbonate buffer (pH 9.6) and 75 μL/well was coated in a 96-well ELISA plate and stored at 4 °C overnight. Then the coating anti-IgM Ab is dumped out and the plates were blocked with blocking buffer containing phosphate buffer saline (PBS) with 5% non-fat dry milk and 0.5% Tween 20 for half an hour. The plates were then washed three to six times before adding the test sera. The samples and the control sera were then diluted 1:40 in the wash buffer containing PBS with 0.05% Tween 20, pH 7.2. Fifty microliters of the diluted samples were added to each well and incubated at 37 °C for 1 h in a humidified chamber. The plates were then washed and the diluted DENV antigen mix was added to each well. The four DENV antigens (TV003 vaccine viruses) were mixed equivalently in wash buffer and 50 μL of the diluted antigen was added

to each well. For each serum sample and time point, normal mock supernatant from Vero-81 was used to serve as antigen negative control to pair with DENV antigen mix. The wells were incubated with the antigen mix at 4 °C in a humidified chamber. The plates were then washed and horse-radish peroxidase (HRP) conjugated PanFlavi monoclonal Ab 6B6C-1(InBios Cat#500510D) was added. The HRP conjugated detecting Ab was diluted 1:1000 in blocking buffer and 50 μL/well was added and incubated for 1 h at 37 °C in humidified chamber. The plates were washed and 75 μL/well of the TMB substrate (enhanced K-Blue TMB substrate, Neogen Cat# 308175) was added. The plates were incubated at room temperature for 30 min or until the positive control turned very dark blue while the negative controls were still negative. The reaction was stopped by adding 50 μL of 1 N H₂SO₄. The blue color changed to yellow and the plates were read at 450 nm. For positive signals, the signal with viral antigen should be 2X greater than signal with normal antigen (internal control); if not, the result is inconclusive. If the P/N value calculated as in Eq. (1) is <2, the result is negative, if P/N value is ≥2 and less than 3 then the result is considered equivocal; if P/N ≥ 3 then the result is positive.

$$\frac{P}{N} = \frac{\text{Mean OD of sample or positive control reacted with viral antigen}(P)}{\text{Mean OD of the normal human serum reacted with viral antigen}(N)} \quad (1)$$

**Neutralization assay.** We used the focus reduction neutralization test (FRNT) to perform the neutralization assays[50,51]. 96-well plates were coated with $2 \times 10^4$ vero-81 cells/well and incubated at 37 °C for 24 h. Neutralization titers were determined by threefold serial dilutions of human samples mixed with 50–100 focus-forming units per well in DMEM-F12 supplemented with 2% FBS. The Ab-virus mixtures were incubated at 37 °C for 1 h before transferring to the confluent 96-well plates. Following an additional 1 h incubation at 37 °C, the monolayers were overlaid with Opti-MEM (Gibco, Grand Island, NY) containing 2% FBS and 1% (wt/vol) car-boxymethyl cellulose (Sigma, St. Louis, MO).

Infected plates were incubated for 2–3 days at 37 °C with 5% CO₂, at which time they were fixed with paraformaldehyde, permeabilized, blocked with 5% non-fat dried milk in permealization buffer, incubated with 4G2 and 2H2 primary mouse mAbs at a dilution of 1:300 each in permeabilization buffer followed by secondary horse-radish peroxidase (HRP)-conjugated anti-mouse IgG (KPL) at a dilution of 1:800 in permeabilization buffer, washed again and developed with TrueBlue peroxidase substate (KPL). We calculated IC₅₀ values by using the sigmoidal dose response (variable slope) equation of Prism 6 (GraphPad Software, San Diego, CA, USA). Reported values were required to have an $R^2 > 0.75$, a hill slope >0.5, and an IC₅₀ within the range of the dilutions.

Using the neutralization 50% (Neut₅₀) titers the percentage of TS nAbs against each serotype were calculated using the following Eq. (2).

$$\%\text{TS nAbs} = \frac{\text{Neut 50 titer (heterologous depletion)} - \text{Neut50 titer(homologous depletion)}}{\text{Neut50 titer (BSAdepeltion)} - \text{Neut50 titer (homologous depletion)}} \times 100$$

$$(2)$$

**rDENVs used for epitope mapping studies.** Recombinant viruses were constructed using a four-cDNA cloning strategy, the same strategy used to create wt DENV infectious clones. The DENV1 (West Pac 74), DENV2 (S-16803), DENV3 (Sri Lanka 89), and DENV4 (Sri Lanka 92) strains were used in the present study. Epitope transplanted recombinant DENVs; DENV2 with the DENV1-1F4 epitope from DENV1 (rDENV2/1), DENV 4 with the E Domain III from DENV2 (DENV4/2), DENV 4 with the DENV3-5J7 epitope from DENV3 (DENV4/3) and DENV2 with the DENV4-126 and DENV4-131-neutralizing epitopes (DENV2/4) were used in this study (Table S3). The full-length cDNA was transcribed into genome-length RNAs using T7 polymerase, as previously described by our group[35,52,53]. Recombinant genome length RNA was electroporated into C6/36 cells, and cell culture supernatant containing viable virus was harvested. Virus was then passaged twice on C6/36 cells, centrifuged to remove cellular debris, and stored at −80 °C. Passage 3 represents our working stock.

The generated recombinant rDENV2/4 virus was tested using a panel of DENV2 (DENV2-2D22, DENV2-3F9, DENV2-4J23) and DENV4 type-specific Abs (DENV-5H2, DENV4-126, DENV4-131) and broadly cross-neutralizing envelope dimer epitope-specific (EDE) antibodies (EDE1-C8, EDE2-B7) at a range of concentrations ranging from (10–0.0001 μg/mL) performing threefold dilutions starting from 10 μg/mL in neutralization assays to demonstrate structural and functional transplantation of the epitopes. Additionally, DENV4 and DENV2 primary infection polyclonal sera were also tested against this virus to check if the transplanted epitopes are targets of the DENV4-specific nAbs in neutralization assays.

**Statistical analyses.** Nonparametric Friedman Dunn's multiple comparisons tests were performed to analyze the differences in mean percentage of nAbs, TS responses and absolute TS neutralization titer against the four serotypes. Similarly, nonparametric Friedman Dunn's multiple comparisons test was also used for the epitope mapping data to determine the loss or gain in neutralization to the recombinant chimeric viruses compared to the wild type parental backbone viruses. We performed an exploratory statistical analysis to evaluate the relationship between pre-challenge nAb titer and boost in DENV2 nAb titer post-challenge. We evaluated the correlation between predictor and outcome variables using Pearson's product-moment correlation. We estimated the odds ratio of ≥2-fold boost in

DENV2 nAb titer for those with predictor nAb levels above vs. at or below a specified threshold. We used the mosaic package in R to perform the odds ratio analysis. We evaluated threshold values corresponding to each decile (20th to 80th percentiles) for each predictor variable. As this is an exploratory analysis, we present the threshold for the decile with the most significant relationship with fold-boost. The threshold corresponding to the 20th percentile had the lowest p-value for the predictors: Total DENV2 nAb, TS DENV2 nAb, and % of TS DENV2 nAb. The threshold corresponding to the 40th percentile was most significant for the predictor Geometric mean of nAb titers to all four DENV serotypes.

**Reporting summary**. Further information on research design is available in the Nature Research Reporting Summary linked to this article.

## Data availability

The authors declare that all main data generated or analyzed during this study supporting the findings are available within the article and its supplementary information files. Extra data that support the findings of this study are available from the corresponding author upon reasonable request. The source data underlying the figures are provided as a source data file with this paper. The full detailed study or the clinical protocol for the clinical trial is provided as a separate file. Pages 35–37 in the study protocol describe the key objectives of the clinical trial that were assessed to provide the pre-specified outcomes reported in this manuscript. Source data are provided with this paper.

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

## Acknowledgements

We thank the study volunteers and clinical staff at the Johns Hopkins Center for Immunization Research and the University of Vermont Vaccine Testing Center. Funding for this work was provided by the Bill and Melinda Gates Foundation and the NIH, National Institute of allergy and Infectious Diseases (Grant numbers AI107731 and AI106695) and UO1AI141997. Samples were obtained from a clinical trial funded by contract HHSN272200900010C of the Intramural Research Program of the NIH, NIAID.

## Author contributions

U.K.N., J.S., M.D., and B.P. conducted the experiments, analyzed the data. A.M.D. and R.S.B. conceived and designed experiments. K.K.P., B.D.K., S.A.D., A.P.D., and S.S.W. provided the specimens. L.K. performed the correlate analysis. U.K.N., A.M.D., R.S.B. wrote the manuscript. All authors participated in the editorial process and approved the manuscript.

## Competing interests

A.M.D. has consulted on Dengue vaccine for Takeda vaccines, Sanofi Pasteur, GSK, and Merck Pharmaceuticals and also an inventor in patents related to Dengue vaccines. R.S.B. has consulted on Dengue vaccines for Takeda Vaccines and Sanofi Pasteur and is also an inventor in patents related to Dengue vaccines. All other authors report no potential conflicts of interest.
