## [Peer Review File · Nature Communications]

Reviewer #1 (Remarks to the Author):

This is a sophisticated study on the attributes of neutralizing antibodies circulating 6 months after vaccination of 21 seronegative adults with a single dose of a tetravalent live-attenuated dengue vaccine. This is followed by a further study of the anamnestic neutralizing antibody responses observed after challenge with a single dose of live-attenuated dengue 2 virus. Based upon absorption tests, after a single dose of vaccine 13 DENV 1; 16 DENV 2; 18 DENV 3 and all 21 DENV 4 vaccinees circulated type specific neutralizing antibodies. Using novel genetic transplant methods the authors determined that antibodies bearing wild-type DENV neutralizing epitopes for each of the 4 DENVs circulated after a single dose of tetravalent vaccine. Novel to this experiment is the detection of low levels of IgM DENV antibodies post live virus challenge in a small number of subjects in the absence of viremia. Some, but not all had low levels of DENV 1 type specific neutralizing antibodies prior to challenge. This raises the question of whether there might have been some replication of the DENV 2 challenge virus.

Specific comments:

Line 54. The final sentence needs to be stated correctly. Firstly, the word “correlate” appears twice. Importantly, without vaccine test results from a phase 3 study, it is not possible to generalize to the conclusion that the vaccine under study is protective. The word “may” should be introduced.

Line 88. The statement in this sentence is an observation. There is no evidence linking unbalanced replication of vaccine viruses to failure of Dengvaxia to protect against dengue virus infection or to explain enhanced disease in vaccinated seronegatives.

Supplement: What is the reason why the PRNT data for dengue 2, column 2 in table S 1 differ from data in column 1 in table S 4?

Table S 4: The authors observed a 3 – 8-fold increase in DENV 2 neutralizing antibody titers in 5 vaccinees on day 270, 90 days after administering challenge virus. The reviewer wonders if the boost observed in DENV 2 PRNT might be heterotypic antibodies related to low levels of DENV 1 PRNT in sera # 2 and 3, for example? Were boosts observed in DENV 1 PRNTs after challenge? It is not clear when and how long after challenge the IgM antibodies were detected. Are the authors aware that a rapid decline in anamnestic antibody titers was observed in monotypic dengue-immune monkeys challenged with homologous virus while heterologous challenged animals produced a sustained elevation of antibody responses?¹ It would be interesting to know the PRNT titers in these animals 6 months after challenge.

1. Halstead SB, Casals J, Shotwell H, Palumbo N. Studies on the immunization of monkeys against dengue. 1. protection derived from single and sequential virus infections. *AmJTropMedHyg* 1973; 22(3): 365-74.

Reviewer #2 (Remarks to the Author):

Nature Communications MS review

This manuscript by Nivarthi, et al., describes the neutralizing antibody responses induced by TV003 (a tetravalent dengue virus vaccine candidate). The authors aimed to assess whether TV003 elicited balanced and serotype-specific neutralizing antibody responses to all 4 dengue virus serotypes. They demonstrate that immunization with TV003 resulted in neutralizing antibody production against 3 or 4 serotypes in 76% of the volunteers and that following challenge with an attenuated dengue virus 2 strain, 16/21 volunteers had sterilizing immunity against DENV-2.

Major comments:

Figure 1 and corresponding text indicate that only DENV-2 depletion was performed to measure TS nAb against DENV-1, 3, and 4. In the Methods section (lines 421-423), it states that “A protocol using a different depletion condition... would be expected to yield similar results”. Please provide data or a reference to support this claim.

Minor comments:

There is inconsistency in the definition of Ab boost. Methods line 349 indicates ≥ 4 -fold rise in titer, while the text (lines 232, 240, 246, 249...) states ≥ 2 -fold rise in titer.

Line 462-463: Should this read “if P/N value is ≥ 2 and less than 3 then the result is considered equivocal...”?

Supplementary Figure 3: “Cor” and p-values are not shown in the figure.

Response to reviewer's comments

Response to Reviewer Comments

Reviewer#1:

- Line 54. The final sentence needs to be stated correctly. Firstly, the word “correlate” appears twice. Importantly, without vaccine test results from a phase 3 study, it is not possible to generalize to the conclusion that the vaccine under study is protective. The word “may” should be introduced.

Response: We agree with the reviewer and now have removed the word “correlate” twice and introduce the word “may” as suggested. The changes made are highlighted in the main text of the manuscript in lines 54-56 on page 3. The new sentence is rewritten as below.

“Our results indicate that nAbs to TS epitopes on each serotype may be a better correlate than total levels of nAbs currently used for guiding DENV vaccine development”.

- Line 88. The statement in this sentence is an observation. There is no evidence linking unbalanced replication of vaccine viruses to failure of Dengvaxia to protect against dengue virus infection or to explain enhanced disease in vaccinated sero negatives.

Response: We disagree with the reviewer's comment that there is no evidence linking unbalanced replication to vaccine efficacy. For Dengvaxia, several publications have established that the DENV4 component replicates to higher levels than the other three components in humans and animals (Guirakhoo et.al., *Virology*, 2002; Guy et.al., *Am J Trop Med Hyg*, 2009; Osorio et.al., *Am J Trop Med Hyg*, 2011; Morrison et.al., *J Infect Dis*, 2010; Dayan et.al., *Hum Vaccin Immunother*, 2014; Torresi et.al., *J Infect Dis*, 2017). This replication pattern is correlated with most baseline seronegative subjects developing DENV4 TS nAbs and lower levels of DENV1, 2 and 3 CR nAbs after receiving Dengvaxia (Henein et.al., *J Infect Dis*, 2017). Finally, in Sanofi's clinical trials, while overall vaccine efficacy was low, serotype-specific efficacy against DENV4 was high even after 5-6 years of follow up (Sridhar et. al. *N Engl J Med*, 2018; and Gustavo et.al., *Vaccine*, 2020). Moreover, in the Takeda vaccine the DENV2 component replicates to the highest titer compared to the other three serotypes. Most seronegative vaccinated individuals develop high titers of DENV2 TS nAbs (Swanstrom et.al., *J infect Dis*, 2018) and vaccine efficacy was highest for DENV2 in Takeda's clinical trial (Biswal et. al., *N Engl J Med* 2020). We agree with the reviewer that further work is needed to better understand how vaccine virus replication is linked to immunogenicity and efficacy. We have revised the text on page 6 in lines 88 to 95 as follows:

“An unbalanced replication of vaccine components was observed in Dengvaxia where the DENV4 component was replication and immunodominant compared to the other three serotypes. Moreover, unlike the DENV1, 2 and 3 components, the DENV4 component of Dengvaxia elicited serotype-specific nAbs in majority of dengue-naïve individuals. In Dengvaxia clinical trials, vaccine efficacy was highest against DENV4. These observations indicate that independent replication of each vaccine component above a threshold is required for protection”.

Supplement:

- What is the reason why the PRNT data for dengue 2, column 2 in table S1 differ from data in column 1 in table S4?

Response: Since we performed the depletion studies with the WHO reference strains, the PRNT data for DENV2 in column2 in table S1 represents the neutralization titers against DENV2 WHO S16803 strain as mentioned in the “Viruses, Cells” subsection of Methods on page 21 from lines 396 to 401. We calculated the boost against NGC in Columns 1, table S4 which is the DENV2 vaccine strain as it is the most relevant strain to use for calculating the boost. nAb boost against DENV2 was also reported against NGC previously (**Kirkpatrick et al; Science Translational Medicine, 2016**). This could explain the subtle differences in neutralization titers against the two strains performed at different times. We have clearly stated it and highlighted now in the Methods section in lines 412-414 on page 21.

- Table S4: The authors observed a 3-8-fold increase in DENV2 neutralizing antibody titers in 5 vaccinees on day 270, 90 days after administering challenge virus. The reviewer wonders if the boost observed in DENV 2 PRNT might be heterotypic antibodies related to low levels of DENV 1 PRNT in sera # 2 and 3, for example? Were boosts observed in DENV 1 PRNTs after challenge? It is not clear when and how long after challenge the IgM antibodies were detected. Are the authors aware that a rapid decline in anamnestic antibody titers was observed in monotypic dengue-immune monkeys challenged with homologous virus while heterologous challenged animals produced a sustained elevation of antibody responses? It would be interesting to know the PRNT titers in these animals 6 months after challenge.

Response: The reviewer raises an interesting point about the quality of the boost after DENV2 challenge. In this study, we only evaluated the boost response against the challenge serotype (DENV2). It will be interesting to see if the boost is broadly serotype-cross reactive or DENV2 specific. Unfortunately, we do not have sufficient sample to test for boosts against the other serotypes. We assessed IgM only at one time point, 28days post challenge as IgM comes up early after infection.

Reviewer #2:

Major comments

- Figure 1 and corresponding text indicate that only DENV-2 depletion was performed to measure TS nAb against DENV-1, 3, and 4. In the Methods section (lines 421-423), it states that “A protocol using a different depletion condition... would be expected to yield similar results”. Please provide data or a reference to support this claim.

Response: We have provided below, data supporting the claim that “a different strategy or protocol to deplete the sera would be expected to yield similar results” (for example a DENV1 depleted sera to estimate TS nAb against DENV2, 3, and 4 yield similar results”).

Experiment 1: We estimated the proportions of serotype specific nAbs in Subject #6, Subject #19 from this study, by performing depletions using BSA control depletion, DENV1 depletion and DENV234 mix depletion condition instead of DENV2 and DENV134 mix depletion condition as reported in the main text. The absolute TS nAb titers and % TS nAbs against each serotype using this strategy (DENV1 and DV234mix depletion) is compared to the originally reported condition (DENV2 and DENV134 mix) in the table below for the two subjects.

Even though the titers won't exactly match as it is a different experiment, the trends mostly are similar using both strategies showing that all four serotypes develop serotype specific nAbs in subject #6.

Subject #19 did not develop serotype specific nAbs against DENV2 and we got similar results using this strategy as well. The table below is now included as Supplementary table 5 in the manuscript and highlighted on page 23 in lines 449 to 450.

Supplementary table5: Comparison of the two depletion strategies (DENV1 and DENV234mix vs DENV2 and DENV134mix) in subjects 6 and 19

Subject	Depletion Strategy	Absolute serotype specific nAb titer >20 (% type-specific neutralizing nAbs against DENV1-4)			
		DENV1	DENV2	DENV3	DENV4
6	DENV2 and DENV134 mix	170(80)	85(80)	571(85)	89(62)
	DENV1 and DENV234 mix	95(57)	157(69)	811(100)	168(61)
19	DENV2 and DENV134 mix	194(100)	0	722(41)	1183(100)
	DENV1 and DENV234 mix	167(88)	0	890(100)	777(88)

Minor comments:

- There is inconsistency in the definition of Ab boost. Methods line 349 indicates ≥ 4 -fold rise in titer, while the text (lines 232, 240, 246, 249...) states ≥ 2 -fold rise in titer.

Response: The boost definition of ≥ 4 -fold rise in Method line 349 represents how boost was defined by Kirkpatrick et al, 2016, *Science Translational Medicine* in their publication (Reference no 11). In our analysis, to understand the correlation between boost and pre challenge neutralizing antibody response based on the data we independently generated, we used a more stringent definition of ≥ 2 -fold rise which represents text in lines 232, 240, 246, 249 in the previous version of the submitted paper. Moreover, in Kirkpatrick et al publication, boost was defined as a ≥ 4 -fold rise in serum neutralizing antibody titer by study day 270 compared with study day 180. In our analysis, we defined boost as ≥ 2 -fold rise in serum neutralizing antibody titer by study day 208 compared to study day 180. To make it clearer, we now separated the Study design, subjects and sera section in Methods into three subsections (**study design of the clinical trial, pre-specified outcomes, sera used in the current study**). We now added our definition of boost in "Sera used in the current study" sub section and highlighted in lines 383-385 on page 20. We also stated this clearly in our "results section on page 13 in lines 235-238.

- Line 462-463: Should this read "if P/N value is ≥ 2 and less than 3 then the result is considered equivocal..."?

Response: Yes, the reviewer is correct and we have changed this now as suggested and highlighted the changes in line 489 on page 25.

- Supplementary Figure 3: "Cor" and p-values are not shown in the figure

Response: We have included the "Cor" and p-values in the figure as highlighted on page 3 of the Supplementary file.

Reviewer #1 (Remarks to the Author):

This is a very important and comprehensive study of neutralizing antibody responses to live-attenuated and chimeric dengue viruses. This is a complex approach to characterizing neutralizing antibody responses and a huge amount of work. More important, because these vaccinated volunteers were challenged with a live dengue 2 virus, the study also includes neutralizing antibody correlates of protection against disease and production of “sterilizing immunity.” A few suggestions to improve the word flow.

Specific comments:

Line 74. After “formulations” insert “designed”

Line 89. After “Dengvaxia” insert a period. Delete the word “where.” Start new sentence with “The.”

Line 93. The authors have not provided evidence that dengue 4 replicated to a higher titer than did other viruses in individuals given Dengvaxia. An immune response sometimes but does not always imply a larger dose of immunogen.

Line 104. Delete “Furthermore” and insert “In addition,”

Line 111. Change to “Sera from 21 individuals were tested”

Line 117. Replace “could” with “can” or “might.”

Line 272. Insert “of protection” after “correlate”

Line 273. It would be a good idea to mention the vaccines” yellow fever, JE, TBE? Are there more?

Reviewer #2 (Remarks to the Author):

The manuscript entitled “A tetravalent live attenuated dengue virus vaccine stimulates balanced immunity to multiple serotypes in humans” by Nivarthi et al has been improved by the revision process. All reviewer comments have been appropriately addressed. The only remaining minor issue is the syntax of the edited sentence “Serum samples from twenty-one dengue-naïve individuals were analyzed 6 months after vaccination who participated under study protocol CIR287 (ClinicalTrials.gov NCT02021968).” Lines 45-47. This sentence should read “Serum samples from twenty-one dengue-naïve individuals who participated under study protocol CIR287 (ClinicalTrials.gov NCT02021968) were analyzed 6 months after vaccination.”

A tetravalent live attenuated dengue virus vaccine stimulates balanced immunity to multiple serotypes in humans

Response to reviewer's comments version 2

Response to Reviewer's Comments

Specific Comments

Reviewer#1:

- Line 74. After “formulations” insert “designed”.

Response: We inserted “designed” after “formulations” as suggested and the change is now highlighted in line 74 of the manuscript.

- Line 89. After “Dengvaxia” insert a period. Delete the word “where.” Start new sentence with “The.”

Response: We made the changes as advised and the changes are now highlighted in line 89 of the manuscript.

- Line 93. The authors have not provided evidence that dengue 4 replicated to a higher titer than did other viruses in individuals given Dengvaxia. An immune response sometimes but does not always imply a larger dose of immunogen.

Response: In the main manuscript we referred to the human studies conducted by Morrison et.al., J Infect Dis, 2010; Dayan et.al., Hum Vaccin Immunother, 2014; Torresi et.al., J Infect Dis, 2017, referenced as 18, 19, 20 where the viremia levels for each serotype following vaccination with Dengvaxia were measured. All the studies showed that the DENV4 viremia was detected most often, followed by DENV3, DENV1 and in majority of the subjects who received the tetravalent vaccine, no DENV2 viremia was detected. For example, studies by Torresi et al., in dengue-naïve adults in Australia to assess replication and excretion of the live attenuated CYD-TDV into biological fluids following vaccination, showed DENV4 viremia was detected in 69.5% of participants, followed by DENV3 (in 19.7% of participants), and DENV1 (in 12.1% of participants) and serotype 2 was not detected. These studies provided evidence of an unbalanced replication among Dengvaxia vaccine viruses in humans.

These three studies were referred in lines 89, 90 in the main manuscript and now highlighted.

- Line 104. Delete “Furthermore” and insert “In addition,”

Response: We removed the word “Furthermore” and inserted “In addition” as advised. The change is now highlighted in line 104 of the manuscript.

- Line 111. Change to “Sera from 21 individuals were tested”

Response: We made the change as advised and the change is now highlighted in line 111 of the manuscript.

- Line 117. Replace “could” with “can” or “might”

Response: *We made the change as advised and the change is now highlighted in line 118 of the manuscript.*

- Line 272. Insert “of protection” after “correlate”

Response: *We made the change as advised and the change is now highlighted in line 272 of the manuscript.*

- Line 273. It would be a good idea to mention the vaccines “yellow fever, JE, TBE? Are there more?”

Response: *We now mentioned the above-mentioned vaccines and the change is now highlighted in lines 273, 274 of the manuscript.*

Reviewer#2 (remarks to the Author):

- The manuscript entitled “A tetravalent live attenuated dengue virus vaccine stimulates balanced immunity to multiple serotypes in humans” by Nivarthi et al has been improved by the revision process. All reviewer comments have been appropriately addressed. The only remaining minor issue is the syntax of the edited sentence “Serum samples from twenty-one dengue-naïve individuals were analyzed 6 months after vaccination who participated under study protocol CIR287 ([ClinicalTrials.gov](https://clinicaltrials.gov) NCT02021968).” Lines 45-47. This sentence should read “Serum samples from twenty-one dengue-naïve individuals who participated under study protocol CIR287 ([ClinicalTrials.gov](https://clinicaltrials.gov) NCT02021968) were analyzed 6 months after vaccination.”

Response: *We now made the change as advised and the change is now highlighted in lines 45-47 of the manuscript.*